# Targeted DNA methylation *in vivo* using an engineered dCas9-MQ1 fusion protein

Yong Lei[1,2,*], Xiaotian Zhang[1,2,*,†], Jianzhong Su[3,*], Mira Jeong[1,2], Michael C. Gundry[1,2], Yung-Hsin Huang[2,4], Yubin Zhou[5], Wei Li[2] & Margaret A. Goodell[1,2,3]

Comprehensive studies have shown that DNA methylation plays vital roles in both loss of pluripotency and governance of the transcriptome during embryogenesis and subsequent developmental processes. Aberrant DNA methylation patterns have been widely observed in tumorigenesis, ageing and neurodegenerative diseases, highlighting the importance of a systematic understanding of DNA methylation and the dynamic changes of methylomes during disease onset and progression. Here we describe a facile and convenient approach for efficient targeted DNA methylation by fusing inactive Cas9 (dCas9) with an engineered prokaryotic DNA methyltransferase MQ1. Our study presents a rapid and efficient strategy to achieve locus-specific cytosine modifications in the genome without obvious impact on global methylation in 24 h. Finally, we demonstrate our tool can induce targeted CpG methylation in mice by zygote microinjection, thereby demonstrating its potential utility in early development.

[1] Department of Molecular and Human Genetics, Center for Cell and Gene Therapy, Baylor College of Medicine, Houston, Texas 77030, USA. [2] Stem Cells and Regenerative Medicine Center, Baylor College of Medicine, Houston, Texas 77030, USA. [3] Dan L. Duncan Cancer Center and Department of Molecular and Cellular Biology, Baylor College of Medicine, Houston, Texas 77030, USA. [4] Department of Developmental Biology, Baylor College of Medicine, Houston, Texas 77030, USA. [5] Center for Translational Cancer Research, Institute of Biosciences and Technology, Department of Medical Physiology, College of Medicine, Texas A&M University, Houston, Texas 77030, USA. * These authors contributed equally to this work. † Present address: Center for Epigenetics, Van Andel Research Institution, Grand Rapids, Michigan 49503, USA. Correspondence and requests for materials should be addressed to M.A.G. (email: Goodell@bcm.edu).

DNA methylation plays a vital role in normal development and its dysregulation is associated with multiple diseases, including cancer[1–3]. Although high promoter DNA methylation is thought to be associated with low gene expression, recently available whole genome methylomes and transcriptomes have implicated DNA methylation in other activities such as controlling transcription factor binding and previously clear correlations between DNA methylation and gene expression have disintegrated[4,5]. The inability to precisely control de novo DNA methylation in mammalian cells hinders our understanding of how DNA methylation at different sites controls downstream effects. Recent efforts in large-scale projects such as the ENCODE[6] and the Roadmap Epigenomics Projects[7] have enabled the identification of numerous tissue- and disease-specific changes in human epigenetic landscapes; however, how DNA methylation specifically regulates gene expression during development and disease progression remains unclear. Current methods of manipulating DNA methylation are primarily based on global inhibition of DNA methyltransferases via small molecule compounds (for example, hypomethylating agents such as Azacitidine and Decitabine), which cause broad epigenetic changes and activation of endogenous retroviruses[8–10]. A lack of technologies for targeted manipulation of DNA methylation has hindered study of the correlation between locus-specific DNA methylation and gene expression. Generating an easy-approached DNA methyltransferase has potential utility for dissecting the role of DNA methylation in multiple biological processes.

Fusion proteins consisting of eukaryotic DNA methyltransferases or hydroxymethylation enzymes and DNA binding proteins, such as zinc finger proteins and transcription activator-like effectors, have been reported to produce targeted DNA modification[11–13]. However, both zinc finger proteins and transcription activator-like effector-based DNA methyltransferase systems require individual design and the construction of encoding plasmids is labour intensive. Compared with these pioneering tools, CRISPR has a unique advantage in multiplex locus engineering and shows negligible impact on methylated DNA[14,15]. dCas9 as a novel DNA binding platform has been applied to study targeted transcriptional reprogramming, histone acetylation and other biological functions[16–18]. Three recent attempts[19–21] to fuse to dCas9 the mammalian de novo DNA methyltransferase 3A (DNMT3A), either full-length or the catalytic domain (CD), showed efficient targeted DNA methylation but generally required a long incubation (several days) in cells to achieve peak efficacy, potentially limiting these tools from contexts where rapid effects are required.

Here we sought a different approach, harnessing a heterologous DNA methyltransferase to dCas9. Although many different prokaryotic DNA methyltransferases have been identified, only a few exclusively methylate CpG dinucleotides. We focused on one derived from Mollicutes spiroplasma (M. Sss1), strain MQ1, which has been well characterized[22]. In particular, MQ1 DNA methyltransferase (hereafter called 'MQ1') is small (386 amino acids) and functions as a highly efficient de novo DNA methyltransferase, making it an appealing candidate for adaption to mammalian cells[22,23].

In this study, we fuse dCas9 with the MQ1 to perform targeted DNA methylation in human cells. To better control MQ1, we generate a mutant form, dCas9-MQ1$^{Q147L}$, which is able to quickly (within 24 h) and efficiently target DNA methylation without obvious off-target effects. We further show that targeted DNA methylation alters CCCTC-binding factor (CTCF) bindings in human cells. Finally, we demonstrate that our tool is applicable to specifically edit DNA methylation in mouse embryos via zygote microinjection.

## Results

**dCas9-MQ1 is an extremely active CpG methyltransferase.** Initially, we designed a dCas9-MQ1 fusion protein using a version of the MQ1 sequence for optimal mammalian expression by replacing the M. Sss1 codons with the optimized human codons. We fused this gene to the 3′-end of the dCas9 coding sequence and included a T2A sequence, to allow coordinated expression of enhanced green fuorescent ptrotein (EGFP) to monitor transfection (Fig. 1a). To test the utility of dCas9-MQ1 for targeted CpG methylation, we selected a CpG island (CGI) near the human HOXA5 gene[24] (Fig. 1b), as it has been shown to be sensitive to DNA methylation when DNMT3A is overexpressed[25]. We designed three single-guide RNAs (sgRNAs) (sg1–3) near the transcription start site (TSS) and an irrelevant guide to lacZ as a control, which were cloned into a red fluorescent protein (RFP)-based vector for selection. A plasmid containing the fusion construct was transfected into human HEK293T cells along with different combinations of sgRNAs plasmids. After 24 h, GFP and RFP double-positive cells were sorted (Supplementary Fig. 1a), and genomic DNA was extracted, bisulfite treated and used as a template for PCR. Methylation patterns for amplicons from each group were assessed using Illumina NextSeq high-through sequencing. As expected, the background level of methylation of this region in untransfected HEK293T cells or those transfected with sgLacZ alone was very low, with an average methylation level around 10% (Fig. 1c). When dCas9-MQ1 was transfected with guide RNA 1 (sg1), DNA methylation across the whole target site was high, with methylation level of ∼70%. We observed an evident methylation gap at the sgRNA-binding site, indicating that dCas9 binding can protect the 20 bp recognition site from methylation (Fig. 1c). Inclusion of guide RNAs 2 and 3 in transfection showed similar results with high DNA methylation and protection at the guide-binding site, although no methylation gap was seen with sg3 as its target lacks a CpG site (Fig. 1c). Groups with multiple sgRNA (sg1–3) showed less methylation activity; possibly due to the limited space between the guides, which may cause interference between DNA-bound dCas9-MQ1 fusion proteins (Supplementary Fig. 1b).

To evaluate off-target effects, we then examined methylation in nearby targets (Target 2 and 3) in the same HOXA5 CGI and in an upstream CGI at the HOXA4 locus (Fig. 1b). Strong nonspecific methylation was observed at both off-target sites (Supplementary Fig. 1c,d). These results indicate that dCas9-MQ1 exhibits very strong but uncontrolled DNA methylation. This uncontrolled methylation was likely to be due to recognition by the MQ1 CD of CpG sites irrespective of dCas9, as high DNA methylation across Target 1 was apparent even in the absence of a specific guide RNA (Supplementary Fig. 1e) or when an irrelevant guide to LacZ (sgLacZ) was included (Fig. 1c). Moreover, there were no obvious changes in methylation ratios between 24 and 48 h post transfection (Supplementary Fig. 1e), demonstrating that 24 h of incubation was sufficient for this de novo methylation system.

**dCas9-MQ1$^{Q147L}$ enables targeted CpG methylation.** To mitigate the off-target effects and improve the specificity of this fusion protein, we sought to reduce the nonspecific activity of MQ1. Previous studies[23] and our own computational modelling (Supplementary Fig. 2a) indicated that mutagenesis of critical amino acids involved in mediating DNA recognition (S317), tuning the DNA duplex binding affinity (Q147) and catalysing the covalent complex formation (C141) might abrogate its binding activity. Thus, we introduced these variants (also shown in Fig. 1a) into the MQ1 domain and co-delivered the point-mutated

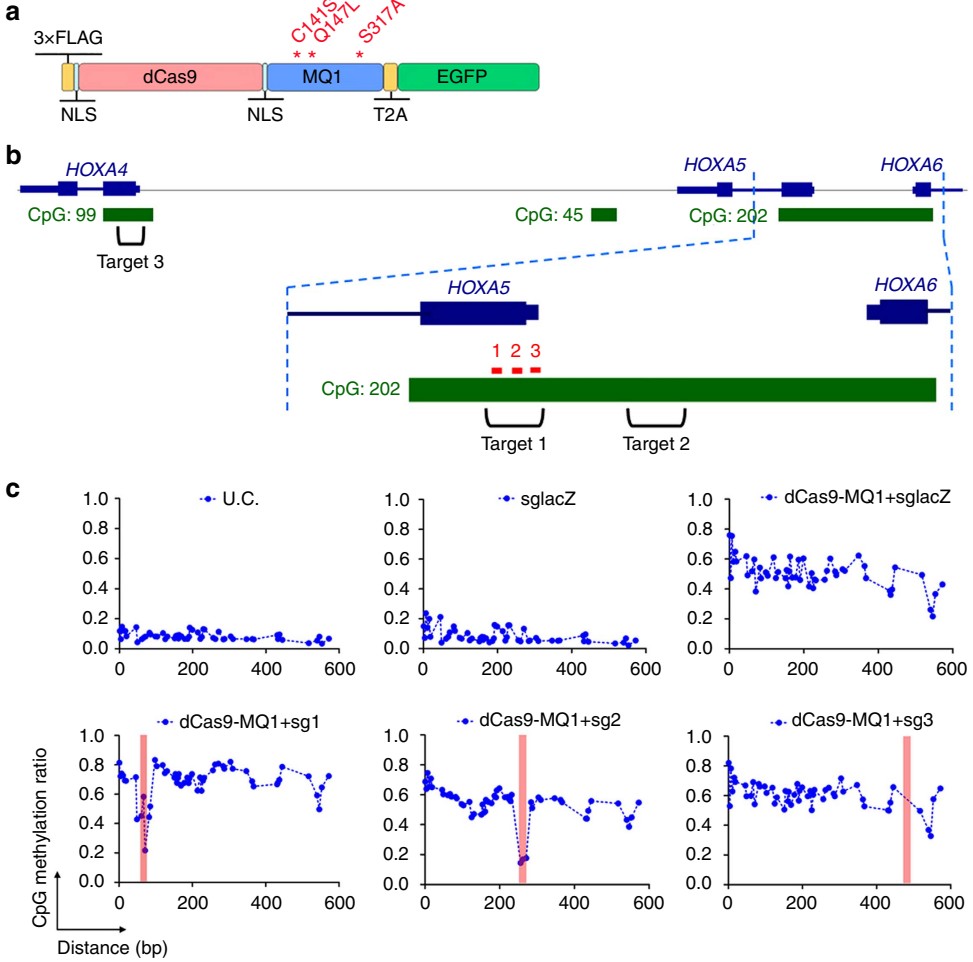

**Figure 1 | dCas9-MQ1 fusion protein induces high *de novo* methylation.** (**a**) Schematic illustration of the domain organization of the dCas9-MQ1-T2A-EGFP expression cassette. $3 \times$ FLAG tag and two NLS signals were inserted at the beginning and the end of dCas9 protein as shown. Mutations made in the current study (C141, Q147 and S317) are shown in red. (**b**) Schematic illustration of the position of *HOXA5* targeted CGI and three *HOXA5* sgRNA-binding sites. *HOXA5* sg1, sg2 and sg3 are located in the Target 1 (on-target sites). Target 2 at CGI 202 (off-target) and Target 3 at CGI 99 (off-target) are also shown. Figure is modified from UCSC Genome Browser human genome (hg38). Not drawn to scale. (**c**) Methylation levels (*y* axis) of individual CpG along the length of CGI (*x* axis) between *HOXA5* and *HOXA6* loci. Red bars indicate the locations of *HOXA5* sgRNA-binding sites. U.C. indicates the untransfected HEK293T control. sgLacZ indicates use of an sgRNA targeting *LacZ*. The 5′ of sgRNA starts from the left side of red bar. Protospacer adjacent motif (PAM) is located on the right side of red bar herein. This direction is applied in all sgRNA sites unless specified.

dCas9-MQ1 and *HOXA5* sgRNAs into HEK293T cells. NextSeq data showed that dCas9-MQ1$^{Q147L}$ caused targeted DNA methylation at *HOXA5* CGI in a highly specific manner (Fig. 2a). dCas9-MQ1$^{Q147L}$ mediated locus-specific methylation at a CpG site ($\sim 60\%$) at about 24 bp downstream of the sgRNA-binding site, whereas minor peaks nearby were also observed. With co-delivery of *HOXA5* sgRNA2, only one weak methylation peak 50 bp upstream of the binding site was observed. When cells were co-transfected with a combinations of sgRNAs (sg1–3), multiple methylation peaks emerged between the sgRNA1- and sgRNA2-binding sites (Fig. 2a). Meanwhile, the variant dCas9-MQ1$^{C141S}$ showed a similar but much weaker methylation effect (Supplementary Fig. 2b). In contrast, the dCas9-MQ1$^{S317A}$ mutant exhibited activity similar to the wild-type MQ1 fusion protein, with hypermethylation of almost every CpG site in the targeted region except for the sgRNA-binding sites. A mutant that combined alterations at both C141 and S317 (hereafter called 'dCas9-dMQ1') lost all methylation capability (Supplementary Fig. 2b). This mutant was used as a loss-of-function control in further experiments. Moreover, we did not observe methylation effects in

dCas9-MQ1$^{Q147L}$ at *HOXA5* target 1 in the absence of guide RNA or presence of sgLacZ (Fig. 2a) and NextSeq results from two off-target regions (Target 2 and 3) confirmed the lack of nonspecific DNA methylation (Supplementary Fig. 2c). Western blotting showed the four dCas9-MQ1 variants had equal expression levels, verifying that the different methylation patterns were not caused by protein expression levels (Fig. 2b). The amount of DNA for the transfection was was determined by dose–response experiments (Supplementary Fig. 3).

Owing to its specific methylation properties, we decided to focus exclusively on the dCas9-MQ1$^{Q147L}$ in further experiments. Previously, two studies showed dCas9-DNMT3a CD reached its maximum methylation activity at day 3–4, or day 7 post transfection[19,20]. To determine the time course for obtaining the best methylation effect, using an available dCas9-DNMT3a CD EGFP plasmid and our tool, we compared CpG methylation activities at the *HOXA5* locus with *HOXA5* sgRNA1 at day 1, day 4 and day 7 post transfection. Cells were harvested via GFP and RFP double-positive sorting. The result (Supplementary Fig. 4) showed that the methylation peak in our tool reaches

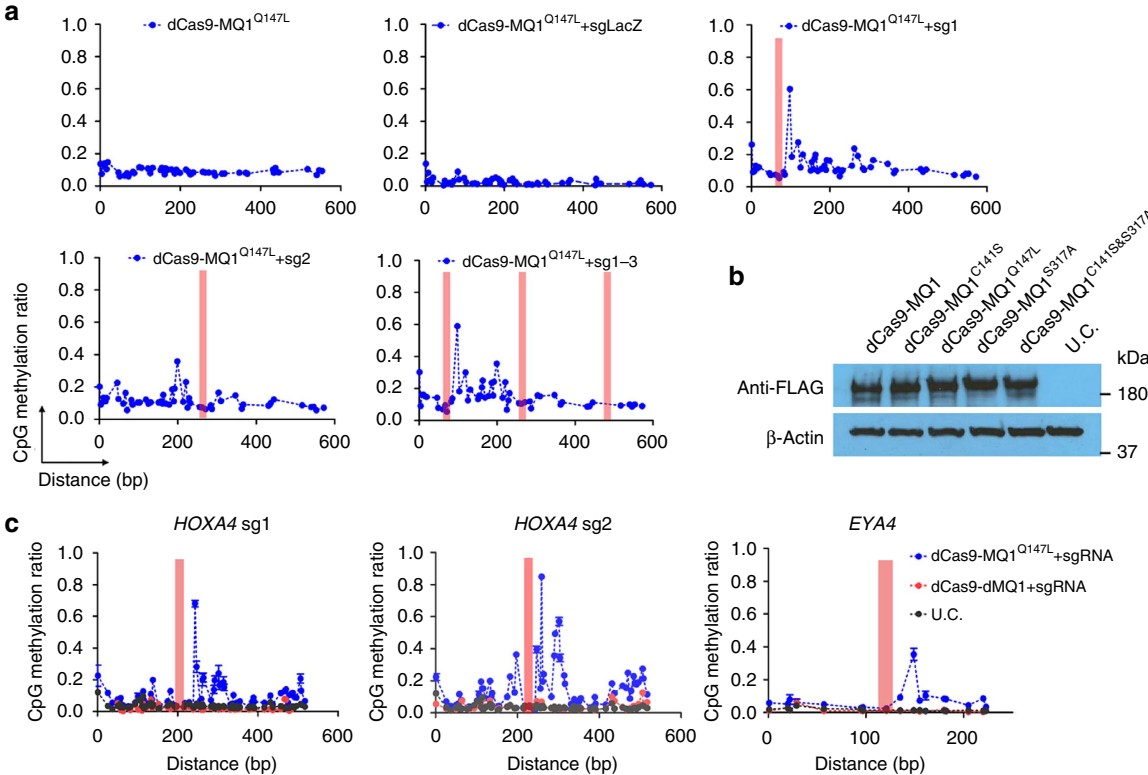

**Figure 2 | dCas9-MQ1[Q147L] induces efficient site-specific CpG methylation.** (**a**) Methylation levels (y axis) at Target 1 (on-target) between *HOXA5* and *HOXA6* loci along the length of the CGI (x axis). Red bars indicate the locations of sgRNA-binding sites and PAM is located on the right side of sgRNA-binding site. The guide RNAs included in the experiment depicted are indicated in each panel, with sg1-3 including all three *HOXA5* Target1 guides. (**b**) Western blotting for protein expression levels of various dCas9-MQ1 mutant fusion proteins in HEK293T cells. Anti-FLAG antibody was utilized and β-actin was selected as internal control. (**c**) DNA methylation analysis as above when guides targeting the *HOXA4* or *EYA4* loci are co-transfected with dCas9-MQ1[Q147L]. Red bars indicate the locations of *HOXA4* and *EYA4* sgRNA-binding sites. The methylation ratio of CpG sites was calculated based on three biological replicates. Error bars represent mean ± s.e.m. of biological triplicates.

about 60% at day 1 and then is sustained at about 30% at both day 4 and day 7. The dCas9-DNMT3a CD tool reaches its peak at day 7, which is consistent with a previous report[20]. These results indicate that dCas9-MQ1[Q147L] induces targeted methylation faster than the available dCas9-DNMT3a-based tool. Moreover, our tool exhibits a relatively focused methylation pattern with a width of about 30 bp. The dCas9-DNMT3a CD tool methylates a much wider region (about 150 bp). These results indicate that the two tools may be optimal for different applications.

When we examined greater distances from the target site, we exclusively detected CpG methylation about 1.5 kb away from the sgRNA binding sites using the dCas9-DNMT3a CD tool, with the methylation ratio increasing in a time-dependent manner (Supplementary Fig. 4). The basal methylation of this region is very low (Supplementary Fig. 1c) and remained at a low level in the samples transfected with our tool.

Using the dCas9-MQ1[Q147L] system, we used three sgRNAs targeting two additional CGIs (*HOXA4* and *EYA4*) on different chromosomes to investigate the consistency of the methylation behaviour. NextSeq results from all three sgRNA targeting regions showed major methylation peaks at 24 or 26 bp downstream from sgRNA-binding sites, as well as some minor peaks nearby (Fig. 2c). Together with the previous methylation study at the *HOXA5* locus, we summarized the methylation profile (Supplementary Fig. 5a) and posit that 20–30 bp may be the most efficient methylation distance downstream of the sgRNA-binding sites, which is probably determined by the spatial structure of the dCas9-MQ1[Q147L] fusion protein. Indeed, the

modelled structure of MQ1 with the linker spans a lateral distance of ∼75∼80 Å, which is comparable to the distance of ∼22–24 bp by assuming an average distance of 3.4 Å between based pairs. Histone modification or chromatin structure may also contribute to shaping DNA methylation patterns. In addition, our tool could be applied in partially methylated regions such as the *GATA2* locus, with evident methylation enhancement observed (Supplementary Fig. 5b).

**dCas9 anchoring impacts transcription factor binding.** Whether targeted methylation affects gene transcription or transcription factor binding required further investigation. Reverse transcription–quantitative PCR (qPCR) was performed using the HEK293T cells at 48 h post co-transfection with dCas9-MQ1[Q147L] and *HOXA5* sgRNAs. Although about a 40% reduction of *HOXA5* gene expression was observed (Supplementary Fig. 6) ($P < 0.001$, in comparison with untreated control, two-tailed t-test), a similar decrease was also seen in the control dCas9-dMQ1 and sgRNA group ($P = 0.21$, two-tailed t-test), which indicated the anchoring of dCas9-based fusion protein hindered transcription initiation or elongation. Then, we applied two sgRNAs targeting transcription factor (TF) SP1-binding sites at cyclin-dependent kinase inhibitor 1 (*CDKN1*) promoter in HEK293T cells for 48 h[26]. Both dCas9-MQ1[Q147L] and the non-functional dCas9-dMQ1 decreased SP1 binding, although targeted methylation only occurred in dCas9-MQ1[Q147L] group (Supplementary Fig. 7). These results demonstrate that the

anchoring of the complex by the dCas9 fusion protein, but not methylation, accounts for the major influence on gene expression and TF binding herein.

**Targeted CpG methylation affects CTCF binding**. To determine whether targeted DNA methylation, irrespective of dCas9 binding, could have an impact on CTCF binding, we applied our tool in K562 cells, targeting CTCF-binding sites (CBS) at three loci that exhibit low DNA methylation (HOXA11, HOXA13 and RUNX1)[6]. After introduction of dCas9-MQ1$^{Q147L}$ and guides, sequencing results showed that methylation was not increased at the HOXA11 locus and only a single methylation peak was gained at the HOXA13 locus. Long repeating sequences after bisulfite conversion shortened PCR amplicons, which hindered investigation of larger regions at HOXA11 and HOXA13 loci. However, dCas9-MQ1$^{Q147L}$ induced CpG methylation in a region of ∼200 bp at the RUNX1 locus with one single sgRNA (Fig. 3a), demonstrating that methylation patterns vary significantly at different loci. To determine whether CTCF binding was affected, cells with the same treatment were analysed by chromatin immunoprecipitation (ChIP) qPCR and ChIP deep sequencing (ChIP-seq) for CTCF binding. Both experimental results consistently demonstrated that binding of CTCF to the targeted site was significantly reduced at the HOXA13 and RUNX1 loci in which we saw increased methylation. CTCF binding was maintained at the HOXA11 locus where methylation was not changed. Furthermore, in all dCas9-dMQ1 control groups, methylation and CTCF binding were unchanged (Fig. 3b,c), supporting the principle that DNA methylation at CBS alters CTCF binding.

We further investigated gene expression after CTCF looping was altered. Owing to the very low expression of the HOXA cluster in K562 cells (Supplementary Fig. 8a), we did not detect expression form HOXA11 and HOXA13 either before or after the methylation change. However, we observed expression increase from RUNX1 transcript variants after methylation of CBS. We illustrated that only the transcript variants (RUNX1_2&3) with the TSS near the targeted CBS were increased. The expression of the long variant (RUNX1_1) remained unchanged (Supplementary Fig. 8b). This result indicates that inducing targeted methylation at CBS near specific TSS could be utilized to alter gene expression.

**Targeted CpG methylation without global methylation**. To exclude the possibility that our system altered global methylation, we performed reduced representation bisulfite sequencing (RRBS) to detect potential global methylation changes at 1, 2 and 4 days post transfection in HEK 293T cells. We analysed ∼72% of global CGIs (20,060 out of 27,719) covered by all experimental groups (Fig. 4a,b, Supplementary Fig. 9 and Supplementary Table 3). Global methylation was maintained at a normal level after 4 days of incubation with dCas9-MQ1$^{Q147L}$ fusion protein compared with untreated control (Fig. 4b). In contrast, wild-type dCas9-MQ1 fusion protein dramatically increased global CpG methylation within 1-day of incubation (Fig. 4a,b and Supplementary Fig. 9).

To further investigate the off-target effects of the dCas9-MQ1$^{Q147L}$ system, we used RRBS to detect methylation status in the cells with dCas9-MQ1$^{Q147L}$ or dCas9-dMQ1 and HOXA5 sgRNA co-delivered. Both groups showed very high correlation with the untreated control; however, outliers were found (Fig. 4c,d). Further analysis showed these outliers were located randomly on chromosomes. Similar outliers were also observed in the comparison between 1- and 4-day incubation in untreated controls (Fig. 4e), suggestive of incomplete bisulfite conversion at

some CpGs in the genome, despite our overall high conversion rations (around 99%). A previous CRISPR/Cas9 study reported the off-target effect at sgRNA similar loci[27]. Based on prediction from the MIT sgRNA design website (http://crispr.mit.edu/), we examined the top eight predicted off-target loci of HOXA5 sg1. Six of them were successfully investigated via bisulfite PCR amplification and deep-sequencing. None of the targeted sites showed unspecific methylation, compared with untranfected controls (Supplementary Fig. 10). Although we cannot exclude other off-target effects not detected, these data indicated high precision and fidelity of our tool. Hence, the dCas9-MQ1$^{Q147L}$ system is a fast and efficient targeted CpG methylation tool without an obvious impact on global methylation.

**In vivo targeted CpG methylation at an imprinted locus**. To test whether our tool could be applied in vivo, we utilized dCas9-MQ1$^{Q147L}$ system to target the differential methylation region of the paternally imprinted locus Igf2/H19 in mice. The overall strategy was to directly inject a combination of plasmids expressing the methylation construct along with multiple sgRNAs into mouse zygotes (Fig. 5a). After embryo transfer, we analysed adult mice for DNA methylation changes at 3 weeks post birth via tail clipping. There are four CBSs (m1–m4) in the murine Igf2/H19 imprinting region[28]. To target these CBS sites, a total of five sgRNAs were applied (Fig. 5b). After birth of the injected mice, we obtained genomic DNA and subsequently performed epigenome typing via bisulfite treatment followed by PCR and then sequencing across the target region. A significant methylation increase was observed between CBS m3 and m4 in the dCas9-MQ1$^{Q147L}$ 5sgRNAs group compared with untreated control mice ($P < 0.0001$, two-tailed $t$-test) (Fig. 5c,d), indicating that multiple CpG sites (in ∼0.5 kb) could be methylated using this sgRNA combination in vivo. sgRNAs targeting m1 and m2 failed to induce methylation, possibly suggesting that some CpG sites are easier to edit, perhaps due to their epigenetic status or chromosome structure. There was no significant methylation change in the dcas9-dMQ1 group, indicating the targeted DNA methylation was not from the recruitment of endogenous DNA methyltransferases (DNMT1, DNMT3A or DNMT3B) or due to spatial occupancy. A previous study reported that full methylation of these four CBSs leads to aberrant IGF2 and H19 expression, which caused birth weight and adult body mass changes[29]. Nevertheless, no significant changes in weight were observed among the groups in our study (Supplementary Fig. 11).

## Discussion

In this study, we have described a novel dCas9-based approach for targeted CpG methylation to endogenous loci in human cells and in mice. By fusing an optimized bacterial DNA methyltransferase, MQ1, to dCas9, we were able to increase methylation levels specifically (over 60%) at three loci (HOXA4, HOXA5 and RUNX1) with very low methylation baselines (Figs 2a,c and 3a). Our tool achieves high CpG methylation within 24 h after transfection, which offers the potential utility of targeted methylation in vivo, especially during embryogenesis. Other recently described tools that fused mammalian DNMT3A with dCas9 required longer incubation times (several days) to achieve maximal[19–21]. Moreover, the methylation patterns introduced by these tools differ. Our MQ1-based tool maintains a relatively centralized methylation pattern with a width of about 30–50 bp. The dCas9-DNMT3a CD tool methylates a much wider region (about 150 bp) and thus may be more suitable for editing large regions.

In general, our dCas9-MQ1$^{Q147L}$ tool achieved the highest methylation activity at CpG sites 20–30 bp downstream from the

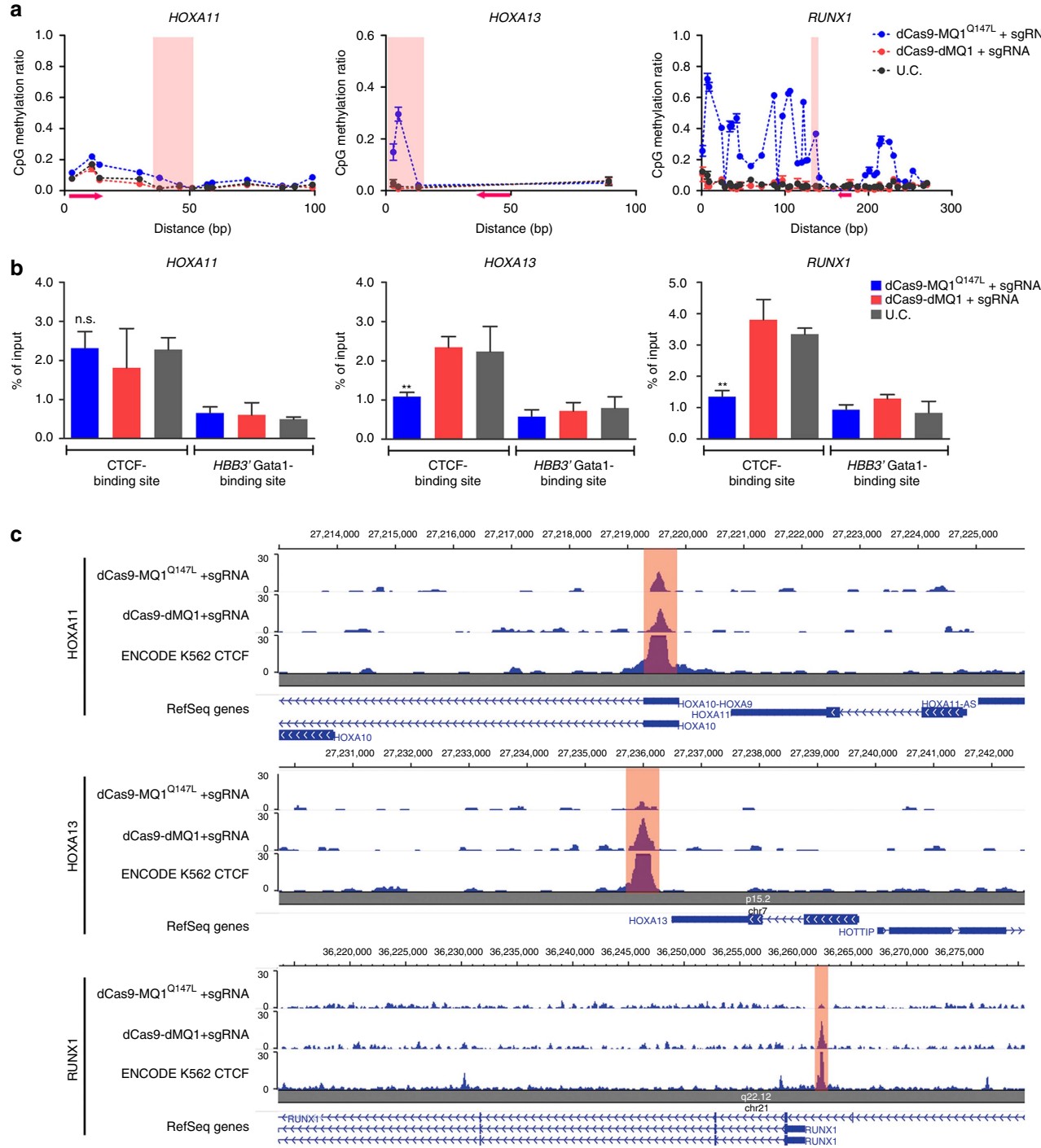

**Figure 3 | Targeted methylation of CBSs to alter CTCF binding.** (**a**) Methylation levels (y axis) of individual CpG along the length of target sites (x axis) at *HOXA11*, *HOXA13* and *RUNX1*. Red arrows indicate the locations of sgRNA-binding sites and towards PAM. Red shadow indicates CBSs. Error bars represent mean ± s.e.m. of biological triplicates. (**b**) Anti-CTCF ChIP was performed using cells in **a** followed by qPCR analysis. Transcription factor GATA1 binding site at the 3′-untranslated region of haemoglobin subunit beta (HBB) was selected as negative control of ChIP-qPCR. Error bars represent mean ± s.e.m. of experimental triplicates. **P < 0.01 (two-tailed t-test). n.s., no significance. (**c**) Anti-CTCF ChIP-seq was performed using cells in **a** followed by Nextseq analysis. Red shadows represent the targeted CBSs in human K562 cells. The reads for each sample was normalized into 20 M via DANPOS software. ENCODE K562 CTCF binding data were applied.

sgRNA-binding site (Fig. 2a,c), consistent with the spatial structure of MQ1. Lack of ideally spaced CpG sites near the sgRNA leads to little or no methylation nearby, such as observed with guides sg2 and sg3 at the *HOXA5* locus. However, the methylation patterns induced by dCas9-MQ1^{Q147L} apparently differed from site to site. In both HEK293T and K562 cells, single or major methylation peaks were obtained at *HOXA* genes, which are located in an unmethylated canyon. Methylation patterns at *CDKN1*, *RUNX1* and *Igf2/H19* were more extended (Figs 3a and 5c, and Supplementary Fig. 7a), suggesting other factors such as

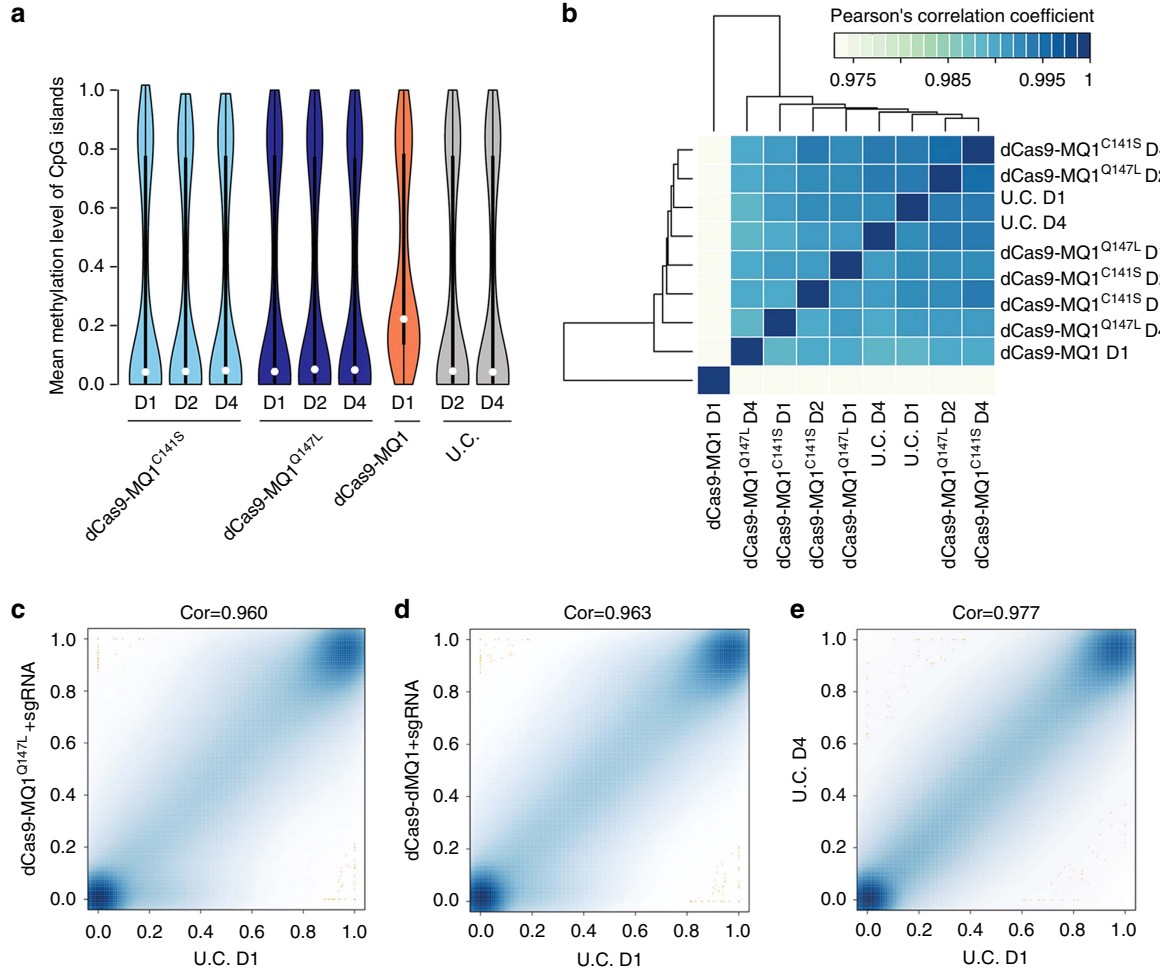

**Figure 4 | Genome-wide CpG methylation analysis in the presence of dCas9-MQ1 variants.** (**a**) RRBS analysis was used. Statistics of mean methylation levels using the indicated mutants without sgRNAs. By comparing the three dCas9-MQ1[Q147L] groups and two untreated control groups, there is a significant increase of DNA methylation in dCas9-MQ1 sample ($P < 1.0E - 130$, two-tailed $t$-test). D1, 1day of incubation; D2, 2 days; D4, 4 days. (**b**) Clustered heat maps for Pearson's correlation for all pair-wise comparisons of RRBS data. About 72% of genome-wide CGIs (20,060 out of 27,719) covered by all groups were analysed. (**c–e**) Methylation ratio density for pair-wise comparisons of RRBS data with sgRNA co-delivered. Cor, correlation coefficient; D1, 1day of incubation; D2, 2 days. U.C., untreated control.

histone modifications or chromatin structure also influence methylation results. These factors may also explain the inability of dCas9-MQ1[Q147L] to methylate the CpG sites at *HOXA11* in human cells, and the *Igf2/H19* m1 and m2 loci in mice. Similar results with regard to the inability to methylate specific targets were also observed using the DNMT3a-based tool[20], suggesting further study regarding the broader influences of DNA methylation activity and dCas9 tools is warranted.

Our data also showed that the dCas9-MQ1 protein strongly prevents the methyltransferase domain from accessing the sgRNA-binding sites. The fusion protein was guided and anchored during the DNA methylation process, resulting in a deep unmethylated gap (Fig. 1c and Supplementary Fig. 2b). The anchoring effect also interferes with transcription elongation and transcription factor binding, which is consistent with previous studies[17,20,30]. However, in the case of targeting the methyltransferase to CTCF sites, we demonstrated that alteration of CpG methylation, rather than protein exclusion, was the major influence on CTCF binding[21] (Fig. 3). Indeed, due to its high specificity, this tool may have a unique advantage in targeting small elements such as CTCF sites, which can then be used to probe gene expression effects via adjacent loops alteration

(Supplementary Fig. 8). Repression of gene expression may be optimally achieved via dCas9-based repressors or CRISPRi[31] technologies, because fully and precisely methylating whole promoter regions remains a challenge.

Off-target effects continue to be a concern for existing Cas9-based tools. With our optimized MQ1[Q147L] tool, we did not detect unspecific effects in genome-wide deep sequencing analysis, sgRNA similar sites or extended regions near the intended target sites. These results contrast with the DNMT3a-based tool, which induced unspecific activities in a previous publication[19] as well as in our study (Supplementary Fig. 4). Nevertheless, we cannot entirely exclude any off-target effects.

Recently, a number of Cas9-DNA methylation tools have been developed. A recent study fused dCas9 to DNMT3a–DNMT3L and achieved efficient and widespread effects[32]. Two other studies showed targeted CpG methylation could induce stable silencing of endogenous genes and specific alteration of CTCF looping[21,33]. To widen potential applications, we tested our tool *in vivo*, co-delivering five sgRNA with dCas9-MQ1[Q147L]-encoding plasmids into mouse zygotes by microinjection. Methylation of a ∼0.5 kb region around *Igf2/H19* CBSs m3 and m4 was

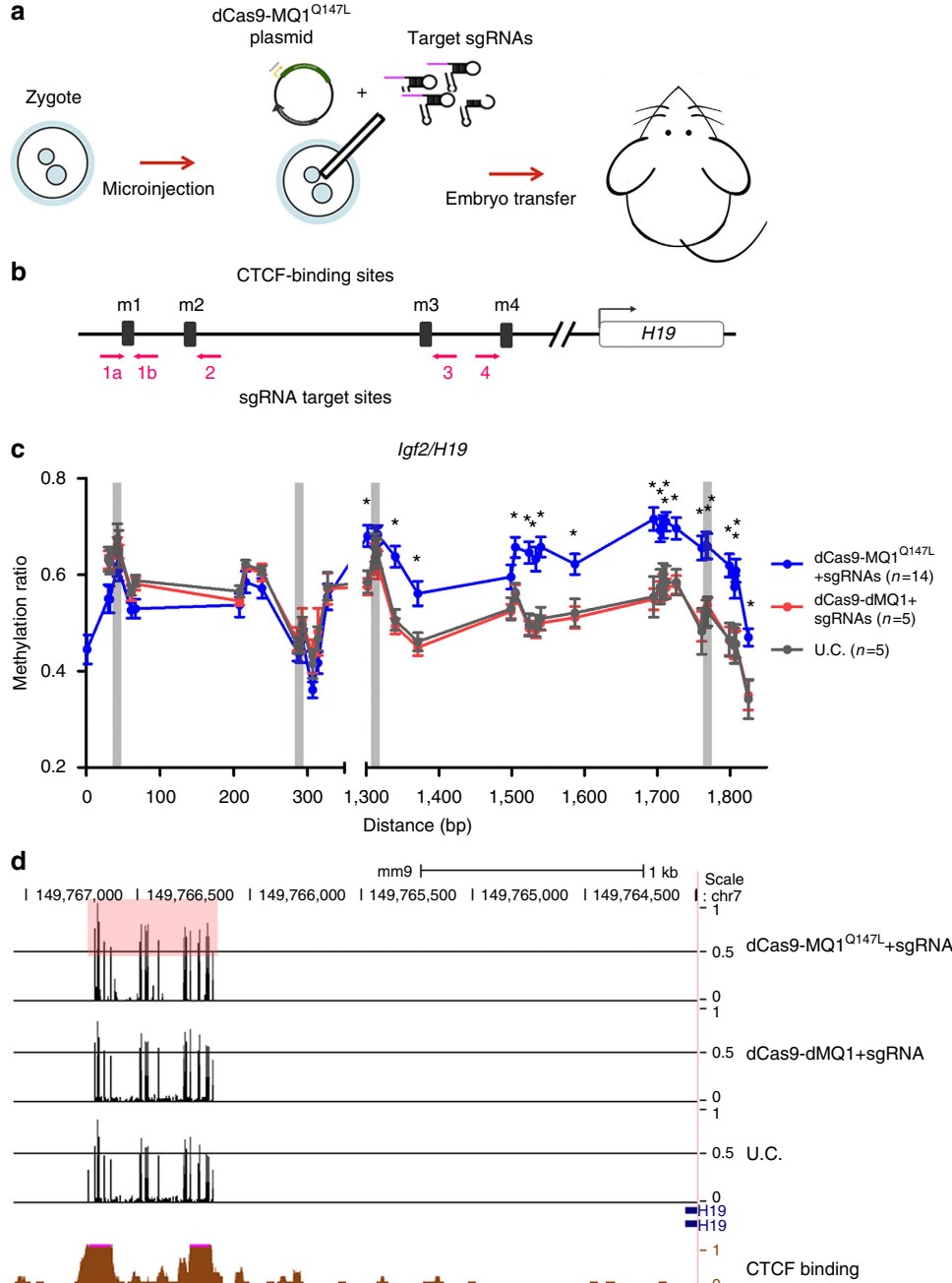

**Figure 5 | Targeted *in vivo* CpG methylation by dCas9-MQ1^Q147L and sgRNAs microinjection in mice zygotes.** (**a**) Schematic diagram illustrating the experimental procedure for targeted CpG methylation *in vivo*. dCas9-MQ1^Q147L encoding plasmid and sgRNAs combination were injected into mice zygotes. (**b**) Schematic representation of targeting CBSs. *Igf2/H19* CBSs were showed as m1, m2, m3 and m4. Arrows indicate the locations of sgRNA-binding sites and towards PAM. Not draw to scale. (**c**) Murine epigenome typing results of targeted methylation regions. Methylation levels (*y* axis) of individual CpG along the length of target sites (*x* axis) at *Igf2/H19* locus among groups. *n* indicates the number of mice used for epigenome typing. *Represents a significant difference between dCas9-MQ1^Q147L + 5sgRNAs group and untreated control (two-tailed *t*-test). Error bars represent mean ± s.e.m. of methylation levels of mice in groups. (**d**) Methylation status at the region around *Igf2/H19* CBSs m3 and m4 from a random selected mouse in each group. Red shadow box indicates significant changed CpG sites.

significantly increased (Fig. 5c,d). These results indicate that targeted methylation is heritable during cell proliferation and differentiation for at least 3 weeks. To our knowledge, these data are the first to demonstrate *in vivo* epigenome editing via murine zygote injection.

Taken together, our dCas9-MQ1^Q147L system is a convenient tool to introduce site-specific DNA methylation with high activity and specificity. It is also a straightforward approach for *in vivo*

DNA methylation editing, suggesting its broad applications for investigating gene dysregulation in various disease contexts.

## Methods

**Cell lines.** HEK293T (ATCC CRL-3216) and K562 (ATCC CCL-243) cells were purchased from the American Tissue Collection Center (ATCC, Manassas, VA) with validated cell identity and eliminated of mycoplasma contamination. HEK293T cells were cultured in DMEM medium supplemented with 10% fetal

bovine serum and 1% penicillin/streptomycin, whereas K562 cells were grown and maintained in RPMI (Invitrogen) containing 15% fetal bovine serum and maintained at 37 °C and 5% $CO_2$. Transfections were performed in six-well plates with $0.6 \times 10^6$ HEK293T cells per well by using 1 µg of respective dCas9-MQ1/dCas9-MQ1 mutant expression vectors with or without 2 µg of individual or pooled sgRNA expression vectors with Lipofectamine 3000 (Life Technologies) as per the manufacturer's instruction. Plasmid pmax-GFP was used as transfection control. K562 cells are electroporated by NEON electroporation kit (Invitrogen) under the condition recommended by the manufacturer (1450 V, 10 ms, 4 pulses). In total, 2 µg of respective dCas9-MQ1/dCas9-MQ1 mutant expression vectors with or without 2 µg of individual or pooled sgRNA expression vectors were electroporated into $1.0 \times 10^6$ K562 cells. Fluorescent microscopy was used to ensure the transfection efficiency in all transfections. Transfected or electroporated cells were analysed and harvested by FACS analysis. GFP (dCas9-MQ1 or its mutants vector) and RFP (sgRNA vector) double-positive selection was applied. Fluorescein isothiocyanate and allophycocyanin (APC) were selected for gating strategy.

**Plasmid construction.** The genetic codon for DNA methyltransferase domain MQ1 was humanized with the minigene synthesized by GENEWIZ (South Plainfield, NJ) and subsequently inserted into the vector pLV-hUbC-dCas9-T2A-GFP (Addgene, 53191) via an NheI (NEB) restriction site. The expression cassette dCas9-MQ1-T2A-GFP was then transferred to the pcDNA3.1 backbone (ThermoFisher) between the XbaI (NEB) and AgeI-HF (NEB) restriction sites, and the new vector was named as pcDNA3.1-dCas9-MQ1. Next, the plasmids: pcDNA3.1-dCas9-MQ1$^{C141S}$, pcDNA3.1-dCas9-MQ1$^{Q147L}$, pcDNA3.1-dCas9-MQ1$^{S317A}$ and pcDNA3.1-dCas9-dMQ1 were created by using QuikChange II XL Site-Directed Mutagenesis Kit (Agilent Technologies as standard instruction). The plasmids for mouse zygote injection (pLV-dCas9-MQ1$^{Q147L}$ or pLV-dCas9-dMQ1) were also created using identical strategy. sgRNAs (see also Supplementary Table 1) were inserted into pLKO5.sgRNA.EFS.tRFP657 (Addgene, 57824) via BsmBI (NEB) site[34]. Plasmid dCas9-DNMT3a CD-EGFP was obtained from Addgene (Addgene, 71666). Oligoes used in this study are provided in Supplementary Table 2.

**sgRNA *in vitro* transcription.** The sgRNA *in vitro* transcription strategy was based on our previous study[35]. In brief, the forward oligo containing the protospacer sequences were designed *via* the CRISPRscan algorithm (http://www.crisprscan.org). Universal reverse oligo 5′-AAAAGCACCGACTCGG TGCCACTTTTTCAAGTTGATAACGGACTAGCCTTATTTTAACTTGCTATT TCTAGCTCTAAAAC-3′ and plasmid PX-458 (Addgene, 48138) were applied together to synthesize sgRNA expression cassettes. sgRNAs were *in vitro* transcribed using T7 RNA polymerase (NEB) and purified by RNA Clean & Concentrator Kit (Zymo). Relevant sgRNA sequences are provided in Supplementary Tables 1 and 2.

**Murine zygote injection.** Mouse zygote injections were manipulated by the Genetically Engineered Mouse Core at Baylor College of Medicine. Briefly, 100 ng µl$^{-1}$ sgRNA combinations and 4 ng µl$^{-1}$ encoding plasmid (pLV-dCas9-MQ1$^{Q147L}$ or pLV-dCas9-dMQ1) were injected into C57BL/6 zygote by pronuclear injection. Manipulated embryos were transferred to female mice with peusdo pregancy. All mouse keeping and handling were performed according to Baylor College of Medicine animal guidelines. For the epigenome typing, mouse tail clipping was performed 3 weeks after birth.

**Bisulfite PCR.** Cells were harvested at 24 h post transfection (or longer as indicated). Genomic DNA from sorted cells or digested mouse tail was extracted by PureLink Genomic DNA Mini Kit (Invitrogen) and bisulfite-converted by EpiTect Bisulfite Kit (Qiagen). Bisulfite-treated DNA (50–100 ng) was amplified with gene-specific primer pairs in a touch-down PCR programme. AllPrep DNA/RNA Micro Kit (Qiagen) was used if both genomic DNA and RNA are needed for further studies. Sequences for the primers used in this study are provided in Supplementary Table 2.

**Bisulfited amplicon Nextseq analysis.** PCR amplicons from bisulfite-treated genomic DNA were quantified by Quant-iT PicoGreen dsDNA Reagent (Invitrogen) and diluted to 0.2 ng µl$^{-1}$ for NextSeq library preparation. The NextSeq libraries were prepared by Nextera XT DNA Library Preparation Kit (Illumina) and Nextera XT Index Kit (Illumina) as per the manufacturer's instructions. Library validation was performed by the Genomic and RNA Profiling Core, Baylor College of Medicine. NextSeq was run using NextSeq Reagent Kits (Illumina). Pooled amplicons libraries were sequenced using an Illumina Nextseq machine. In addition, the paired-end reads were aligned to the human genome hg19 or mouse genome mm9 and low-quality sequences were trimmed using the software BSMAP. We achieved average coverage of more than ×2,000 of measured CpGs for each samples. Only CpG covered by at least 100× reads were further computed for their methylation ratio by the bsratio module in software

BSMAP. All samples underwent bisulfite conversion with an efficiency of at least 96% as judged by the conversion of unmethylated non-CpG cytosines.

**ChIP-sequencing (ChIP-qPCR).** The K562 cells at 48 h post electroporation were sorted and crosslinked with 1% formaldehyde at room temperature for 10 min and the reaction was stopped by 0.125 M glycine at room temperature for 5 min. Crosslinked cells were lysed with nuclear lysis buffer and sonicated to 200–500 bp fragments (Bioruptor, Diagenode). Anti-CTCF ChIP-qualified antibodies (4 µg per sample, Cell Signaling, ab70303) or anti-SP1 antibody (4 µg per sample, Santa Cruz, sc-17824) were added to the sonicated chromatin and incubated at 4 °C overnight. Following this, 10 µl of protein A magnetic beads or protein G magnetic beads (Dynal, Invitrogen) previously washed in RIPA buffer were added and incubated for an additional 2 h at 4 °C. The bead:protein complexes were washed three times with RIPA buffer and twice with TE buffer. Following transfer into new 1.5 ml collection tube, genomic DNA was eluted for 2 h at 68 °C in 100 µl Complete Elution Buffer (20 mM Tris pH 7.5, 5 mM EDTA, 50 mM NaCl, 1% SDS, 50 µg ml$^{-1}$ proteinase K) and combined with a second elution of 100 µl Elution Buffer (20 mM Tris pH 7.5, 5 mM EDTA, 50 mM NaCl) for 10 min at 68 °C. ChIPed DNA was purified by MinElute Purification Kit (Qiagen) and eluted in 90 µl elution buffer. Deep sequencing or qPCRs were then performed. The ChIP-seq DNA library was prepared by ThruPLEX DNA-seq Kit (Rubicon Genomics) as per the manufacturer's instructions. ChIP-seq was run using NextSeq Reagent Kits (Illumina). Pooled libraries were sequenced using an Illumina Nextseq machine. The raw sequencing reads of ChIP-seq were mapped to hg19 human genome by Bowtie 2 software at least two mismatches. Multiple mapped reads were removed. The total number of uniquely mapped reads of six samples were normalized to the same 20 millions using the quantile normalization of software DANPOS 2.2.2.

**Real-time PCR (qPCR).** Genomic DNA and relevant RNA was extracted simultaneously from fluorescent sorted cells using AllPrep DNA/RNA Micro Kit (Qiagen) if both DNA and RNA were needed. Total RNA was reverse transcribed using the SuperScript III First Strand Synthesis SuperMix and Random Primer (Invitrogen). Complementary DNA or ChIPed DNA were quantified by using SsoAdvanced universal SYBR Green supermix (Bio-Rad) on a CFX96 Touch Real-Time PCR system (Bio-Rad). Real-time primers sequences were provided in Supplementary Table 2. For each biological replication, at least three technical replicates of real-time PCR assay were done. Statistical significance was determined by comparing experimental samples against the untreated control using a two-tailed *t*-test ($P < 0.05$).

**Western blotting.** HEK293T cells were transfected with various constructs by Lipofectamine 3000 (Life Technologies) following the manufacturer's protocol. Two days post transfection, cells were lysed by RIPA buffer with proteinase inhibitor (Invitrogen) and performed standard immunoblotting analysis. Mouse anti-FLAG (1:2,000, Sigma, F1804) and mouse anti-β-Actin (1:5,000, Santa Cruz, sc-47778) antibodies were used. Orignal blot images are provided in Supplementary Fig. 12.

**Reduced representation bisulfite sequencing.** We generated RRBS libraries based on previously studies[36,37]. Genomic DNA (100–200 ng) was digested with 10 U MspI (NEB), which cuts at CCGG sites in a methylation-insensitive manner. Digested fragments were end repaired, a tailed and ligated to Illumina adaptors. After ligation, 150–700 bp size was selected using gel-cutting methods to cover more CpGs. Size-selected libraries were bisulfite treated using the EpiTect Bisulfite Kit (Qiagen). Ligation efficiency was tested by PCR using TrueSeq primers and Pfu TurboCx hotstart DNA polymerase (Stratagene). After determining the optimized PCR cycle number for each sample, a large-scale PCR reaction (100 µl) was performed. PCR products were sequenced with Illumina HiSeq2000 sequencing systems. Adapter and base quality trimming was performed with Trimgalore for all raw data files using a Phred score threshold of 20. Reads were mapped to the human genome (hg19) with BSMAP version 2.90. Methylation information was summarized over defined regions of interest[38], adding up C and T counts from each covered cytosine and returning a total C and T count for each region. Differential methylation was calculated by Fisher's exact test and P-values were adjusted to q-values using the SLIM method. Differentially methylated regions were selected based on q-value and percent methylation difference cutoffs of 0.05% and 25%, respectively. Coordinates of Refseq gene elements (promoters, 5′ and 3′-untranslated region, coding exons, introns) and CGIs were obtained from the UCSC genome table browser.

**Structural modelling for MQ1.** The model structure of *M. spiroplasma* MQ1 was generated by using the programme I-TASSER[39], which takes a hierarchical approach to predict the three-dimensional protein structure. The best model structure was built using the template of *Mycoplasma penetrans* CpG-specific methyltransferase *M. Mpel* (Protein Data Bank entry: 4DKJ[40]) that shares over 90% sequence homology. The structure was visualized by using the programme PyMol (http://sourceforge.net/projects/pymol/). Owing to the lack of crystal

structure data, the impact of nucleosome structure or DNA topology is not included in this prediction.

**Code availablity.** BSMAP software is available on http://www.mybiosoftware.com/bsmap-2-74-genome-bisulfite-sequence-mapping-program.html. Analysis codes for other NGS data are available upon request.

**Data availability.** Plasmids sequences and target fragment sequences are available upon request. Next generation sequencing data files were deposited and available on Gene Expression Omnibus GSE90692.

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

## Acknowledgements

This work was supported by the NIH (DK092883, CA183252, CA125123, P50CA126752, HG007538, CA193466 and GM112003), the Edward P. Evans Foundation, the Adrienne Helis Malvin Medical Research Foundation, Robert A. Welch Foundation (BE-1913) and by CPRIT.

## Author contributions

Y.L., X.Z. and M.A.G. designed experiments. Y.L., X.Z., J.S., M.J., M.G., Y.H. and Y.Z. conducted experiments and modelling. Y.L., X.Z., J.S., Y.Z., W.L. and M.A.G. analysed the data. Y.L. and M.A.G. wrote and edited paper with contributions from all authors.

## Additional information

**Competing interests:** The authors declare no competing financial interests.

**Publisher's note**: 

