## [Peer review file · Nature Communications]

REVIEWERS' COMMENTS:

Reviewer #1 (Remarks to the Author):

Lei et al. report Cas9-MQ1 tool for targeted methylation claiming the high specificity of the tool as well as the high methylation activity within 24 hours. They fulfilled all the requests asked by me (as well as by reviewer 1), therefore I would recommend the manuscript for publication in Nature methods. They performed side-by-side comparison of their tool with already existing CRISPR/Cas9-DNMT3a. They showed specific advantages of their Cas9-MQ1 tool, such as faster methylation activity and lower non-specific activity.

I would just have one remark. Supplementary figure 4 relies only on a single point for superior activity on day one, mentions biological triplicates and error bars – yet, no error bars are visible in the figure. Since the claim about superior (stronger) activity on day one relies on a single point, it is essential to display, explain and comment the error bars and the underlying data distribution indicating reproducibility.

Reviewer #2 (Remarks to the Author):

In my opinion, the authors have addressed the previous concerns and the new data provided supports that this will be an important tool with multiple applications in diverse fields.

REVIEWERS' COMMENTS:

Reviewer #1 (Remarks to the Author):

Lei et al. report Cas9-MQ1 tool for targeted methylation claiming the high specificity of the tool as well as the high methylation activity within 24 hours. They fulfilled all the requests asked by me (as well as by reviewer 1), therefore I would recommend the manuscript for publication in Nature methods. They performed side-by-side comparison of their tool with already existing CRISPR/Cas9-DNMT3a. They showed specific advantages of their Cas9-MQ1 tool, such as faster methylation activity and lower non-specific activity.

I would just have one remark. Supplementary figure 4 relies only on a single point for superior activity on day one, mentions biological triplicates and error bars – yet, no error bars are visible in the figure. Since the claim about superior (stronger) activity on day one relies on a single point, it is essential to display, explain and comment the error bars and the underlying data distribution indicating reproducibility.

We appreciate reviewer's concern about the error bars. We updated the Supplementary Figure 4 by shrinking the dots and changed the statistic from mean \pm s.e.m. to mean \pm s.d. for better display. Meanwhile, all next generation sequencing (NGS) data files were deposited and publically available on Gene Expression Omnibus (GEO), GSE90692, for the researchers who are interested in the loci in this study.

Reviewer #2 (Remarks to the Author):

In my opinion, the authors have addressed the previous concerns and the new data provided supports that this will be an important tool with multiple applications in diverse fields.

We appreciate reviewer's valuable comments and suggestions during review processes.